# An Al-Biruni Earth Radius Optimization-Based Deep Convolutional Neural Network for Classifying Monkeypox Disease

**DOI:** 10.3390/diagnostics12112892

**Published:** 2022-11-21

**Authors:** Doaa Sami Khafaga, Abdelhameed Ibrahim, El-Sayed M. El-Kenawy, Abdelaziz A. Abdelhamid, Faten Khalid Karim, Seyedali Mirjalili, Nima Khodadadi, Wei Hong Lim, Marwa M. Eid, Mohamed E. Ghoneim

**Affiliations:** 1Department of Computer Sciences, College of Computer and Information Sciences, Princess Nourah bint Abdulrahman University, Riyadh 11671, Saudi Arabia; 2Computer Engineering and Control Systems Department, Faculty of Engineering, Mansoura University, Mansoura 35516, Egypt; 3Department of Communications and Electronics, Delta Higher Institute of Engineering and Technology, Mansoura 35111, Egypt; 4Department of Computer Science, College of Computing and Information Technology, Shaqra University, Shaqra 11961, Saudi Arabia; 5Department of Computer Science, Faculty of Computer and Information Sciences, Ain Shams University, Cairo 11566, Egypt; 6Centre for Artificial Intelligence Research and Optimization, Torrens University Australia, Fortitude Valley, QLD 4006, Australia; 7Yonsei Frontier Lab, Yonsei University, Seoul 03722, Republic of Korea; 8Department of Civil and Environmental Engineering, Florida International University, Miami, FL 33199, USA; 9Faculty of Engineering, Technology and Built Environment, UCSI University, Kuala Lumpur 56000, Malaysia; 10Faculty of Artificial Intelligence, Delta University for Science and Technology, Mansoura 35712, Egypt; 11Department of Mathematical Sciences, Faculty of Applied Science, Umm Al-Qura University, Makkah 21955, Saudi Arabia; 12Faculty of Computers and Artificial Intelligence, Damietta University, Damietta 34511, Egypt

**Keywords:** skin disease, monkeypox infection, Al-Biruni Earth radius, deep learning, meta-heuristic, optimization

## Abstract

Human skin diseases have become increasingly prevalent in recent decades, with millions of individuals in developed countries experiencing monkeypox. Such conditions often carry less obvious but no less devastating risks, including increased vulnerability to monkeypox, cancer, and low self-esteem. Due to the low visual resolution of monkeypox disease images, medical specialists with high-level tools are typically required for a proper diagnosis. The manual diagnosis of monkeypox disease is subjective, time-consuming, and labor-intensive. Therefore, it is necessary to create a computer-aided approach for the automated diagnosis of monkeypox disease. Most research articles on monkeypox disease relied on convolutional neural networks (CNNs) and using classical loss functions, allowing them to pick up discriminative elements in monkeypox images. To enhance this, a novel framework using Al-Biruni Earth radius (BER) optimization-based stochastic fractal search (BERSFS) is proposed to fine-tune the deep CNN layers for classifying monkeypox disease from images. As a first step in the proposed approach, we use deep CNN-based models to learn the embedding of input images in Euclidean space. In the second step, we use an optimized classification model based on the triplet loss function to calculate the distance between pairs of images in Euclidean space and learn features that may be used to distinguish between different cases, including monkeypox cases. The proposed approach uses images of human skin diseases obtained from an African hospital. The experimental results of the study demonstrate the proposed framework’s efficacy, as it outperforms numerous examples of prior research on skin disease problems. On the other hand, statistical experiments with Wilcoxon and analysis of variance (ANOVA) tests are conducted to evaluate the proposed approach in terms of effectiveness and stability. The recorded results confirm the superiority of the proposed method when compared with other optimization algorithms and machine learning models.

## 1. Introduction

Many applications reap the advance of artificial intelligence (AI) and deep learning (DL) methodologies to ease and facilitate the daily lives of humans. Recently, computer systems based on AI reached accuracy up to par with human experts. Applications of AI are spread over a broad spectrum of fields, such as banking transactions, financial affairs, manufacturing, retail, the supply chain, e-commerce, and healthcare [1]. AI has been an important role player in many new business innovations within the technology industry. Its applications include web search engines, video-on-demand recommendations, products of interest, targeted advertising, and self-driving vehicles [2]. The manuscript introduces AI uses in medical science and healthcare services. The most prevalent rule of AI in medical services is accurately detecting any medical issues and attracting the attention of medical teams, enabling them to respond to the threat and thus saving lives.

The application of AI in medical science and health care is spreading rapidly. Common domains of applications in medical image analysis include different fields such as organ localization and abnormality detection [3,4], gene mutation detection [5], and cancer grading [6] and staging [7], and the domains are too numerous to mention and still flourishing. Remarkably, AI methods have recently played a significant role in the COVID-19 pandemic as a tool for diagnosis and severity ranking from medical images [8,9,10]. This success motivates scientists to recruit AI for the emerging threat of a monkeypox outbreak.

Poxviruses are one of the most known vertebrate infectious viruses, including cowpox, vaccinia, variola (smallpox), and monkeypox viruses [11,12]. Human monkeypox is a zoonotic Orthopoxvirus with symptoms similar to poxviruses, including cowpox, vaccinia, and variola (smallpox). The first case report of human infection by the monkeypox virus was in the Democratic Republic of the Congo in 1970. Monkeypox has been considered the most dangerous Orthopoxvirus for human health since that time. In the past, monkeypox was most frequent in the west African countries. However, it has often been reported in urban areas outside the African continent lately [13].

The multinational monkeypox outbreak has choked the world while recovering from the aftershock of the COVID-19 pandemic. As of the end of June 2022, an abrupt and significant surge in the number of cases close to over 4900 human monkeypox instances has occurred which are non-endemic for the disease, spanning across the Western Hemisphere. Transmission has extended beyond cases linked to an initial exposure in Africa to human-to-human spread in each country. Virologists differentiate two variants of the monkeypox virus: the Central Africa clade and the West Africa clade. There is no proper treatment for the monkeypox virus to date [14,15]. Although the mortality rates of monkeypox are considered low [16], early detection empowers precautions with anyone who seems contagious for patient isolation and contact tracing for effective containment of the disease’s spread.

Clinical identification of monkeypox from different pox viruses is difficult. The diagnosis procedure starts with visual observation of skin lesions and the existing history of exposure, followed by testing skin lesions using dermatoscopic images and then confirmation using a polymerase chain reaction (PCR) [17]. This work will be directed to utilizing AI approaches to diagnose monkeypox with a digital skin image classification algorithm. AI techniques based on artificial neural networks (ANNs) could be useful in detecting monkeypox with the help of skin image processing and analysis. Numerous ANN techniques are applied in the diagnosis of skin lesions due to their unique characteristics of learning crucial features from complex data throughout the training process.

Classification models assist physicians in making more accurate diagnoses of skin cancer, skin lesions, and psoriasis [9]. Deep convolutional neural networks (CNNs) [10] have had their capability for general and highly variable tasks across many specific categories revealed [18,19]. Estevan et al. [20] used a deep convolutional neural network (DCNN) to train a model using 129,450 images for binary classification of images into either malignant melanoma or benign tissue. Furthermore, they concluded that the DCNN achieved performance comparable to that of 21 board-certified specialists. In their work, Haenssle et al. [21] compared a CNN’s diagnostic performance against a group of 58 physicians, including 30 experts in the field of dermatology. They trained Google’s Inception v4 CNN architecture and tested its validity using dermatoscopic images and corresponding diagnoses. The CNN’s performance surpassed most dermatologists. Hekler et al. [22] introduced multiclass classification models combining humans and AI to classify images of suspected skin cancer. Combining natural and artificial intelligence achieved superior accuracy compared with the accuracy achieved by artificial or human intelligence alone.

All proposed solutions for lesion detection problems lay on selecting the optimal parameters of the employed model. The selection process is often performed in two phases, namely mathematical modeling and optimization using an optimization method. Optimization is selecting optimal solutions from a pool of competing explanations for the problem under consideration. In general, a multi-dimensional search problem is turned into an optimization problem. In practice, this entails reducing or increasing the quality of the candidate solutions, often represented as a vector in the search space. In recent years, metaheuristic algorithms have been extensively applied among the optimization algorithms for solving classification problems [23]. Their ability to deal with complex, multi-dimensional, and ill-behaved optimization problems and provide satisfying results in a reasonable time has made them the choice of most researchers and practitioners [24,25].

Numerous types of AI have been proposed for use in different applications [26,27]. Combining the attributes of various separate base models creates an ensemble prediction model using the ensemble technique. There are many different methods to put this concept into practice. Resampling the training set is one of the more efficient methods, while others employ other prediction algorithms, adjust different predictive strategy parameters, etc. In order to aggregate the outcomes of each prediction, an ensemble of techniques is utilized [28,29]. Al-Biruni Earth radius (BER) optimization was first proposed recently in [30,31]. When more variables are used in the optimization process, the BER optimization technique performs worse. However, the successful balancing of exploration and exploitation is a considerable benefit. The suggested approach utilizes this benefit by utilizing the BER algorithm. Stochastic fractal search (SFS) [32] has limitations such as a low exploration rate and performance falling when a large number of local optimum solutions are present, despite its simplicity of usage and excellent balance between exploration and exploitation. The BERSFS algorithm is implemented in this study to use the method’s benefits and get around its limitations.

We employ a metaheuristic algorithm for tuning the parameters of the neural network. A novel framework based on Al-Biruni Earth radius (BER) optimization fine-tunes the deep convolutional neural network (DCNN) layers. As a first step in the proposed approach, we use deep CNN-based models to learn the embedding of input images in Euclidean space. In the second step, we use an optimized classification model based on the triplet loss function to calculate the distance between pairs of images in Euclidean space to learn features that may be used to distinguish between different cases, including monkeypox cases. The proposed approach uses images of human facial skin diseases obtained from an African hospital. The experimental results of the study demonstrate the proposed framework’s efficacy, outperforming numerous prior works on skin disease problems. On the other hand, statistical experiments with Wilcoxon and analysis of variance (ANOVA) tests are conducted to evaluate the proposed approach in terms of effectiveness and stability. The recorded results confirm the superiority of the proposed method when compared with other optimization algorithms and machine learning models.

Based on the suggested BERSFS classification model, excellent prediction accuracy of monkeypox disease is achieved. The tested dataset is assessed using a classifier built with the specified BERSFS algorithm. The BERSFS algorithm is tested and compared to PSO [33], Grey Wolf Optimizer (GWO) [34], the Whale Optimization Algorithm (WOA) [35], BER [30], and SFS [32]. Comparisons are made between models, including the decision tree (DT) regressor [36], K-nearest neighbors (K-NN) [37], support vector machine (SVM) [38], and the “BERSFS” algorithm. Additionally, several CNN models including AlexNet [39], VGG19Net [40], GoogLeNet [41], and ResNet-50 [42] are tested.

The key contributions of this work can be summed up as follows:Offer machine learning techniques for predicting monkeypox disease;A new Al-Biruni Earth radius (BER) optimization-based stochastic fractal search (BERSFS) algorithm is suggested;To raise the tested dataset prediction accuracy, a BERSFS-based classifier is created.A comparison of the results of different algorithms to determine which is the most accurate is performed;The Wilcoxon rank-sum and ANOVA tests are used to determine the statistical significance of the BERSFS algorithm;It is possible to generalize and test the BERSFS-based classification algorithm for different kinds of datasets.

Here is how this paper is organized. Section 2 contains a literature review. Section 3 describes the BERSFS algorithm. Section 4 shows the results and analysis. In Section 5, the conclusions and future directions are given.

## 2. Literature Review

Computer-aided methodologies for automatically identifying diseases from images reduce human error and increase detection speed, which helps physicians and radiologists diagnose conditions more easily, thus protecting the patient against medical threats in the early stages. Those methodologies can be categorized into two categories. The first, traditional methodologies, involves methods that apply only image processing techniques. In contrast, the other evolutionary methodologies are methods that apply AI and DL techniques. In the traditional methodologies [43,44,45,46], the most important features used to detect and identify skin diseases are their color and texture information. In addition, the nature of the disease and the severity of the condition play a role in determining the size and shape of each individual lesion. The process of selecting features to use in traditional approaches is time-consuming. Furthermore, identifying the features that are relevant to the problem at hand is essential because it has an impact on the accuracy of the classification algorithm. On the other hand, evolutionary techniques are capable of learning traits in an automatic and effective manner. Utilizing one of the many different image segmentation algorithms available, it divides the image into regions based either on the pixels, the edges, or the regions themselves. In addition, in contrast to conventional approaches, feature extraction makes intelligent decisions regarding filter selection [47].

This section discusses the previously proposed evolutionary methodologies used in classifying skin lesions from a medical image. Arifin et al. [48] proposed a two-stage system to detect skin lesions using color images, starting by applying k-means clustering with color gradient techniques to identify infected skin. Then, they used an ANN to classify diseases. The accuracy of the first stage was 95.99%, while the second stage was 94.016% accurate. A prepossessing phase is added before feature extraction, and those features are used to predict the type of disease [49]. This technique’s accuracy varies depending on the number of extracted features. These features are input to a feedforward ANN for training and testing. The researchers applied this method to nine types of skin diseases with an accuracy of up to 90%. Soliman and AlEnezi [50] proposed resizing the image to extract features using a pretrained CNN. Then, the features were classified using a multi-class SVM. The system detected three types of skin diseases with an accuracy rate of 100% in a very limited dataset.

Esteva et al. [20] described how an image classifier based on a CNN can reach performance that is comparable to that of 21 dermatologists who have received board certification. The three steps of disease diagnosis were devised to determine if a particular skin lesion should be classified as malignant, benign, or non-neoplastic. In addition, a nine-way illness partition was carried out in order to place each lesion that was examined into one of the nine categories that were previously described. In their article [51], Sandeep et al. examined the use of DL-based approaches for the detection of various skin lesions. They came up with a CNN to separate skin lesions into the eight different illness categories. When compared with VGGNet, the classification provided by the suggested approach was completed four times more quickly. In comparison, the proposed system would provide an accuracy of up to 78%, whereas VGGNet only achieved a 71% success rate. It was proposed in [52] that a CNN could perform low-cost image analysis for identification of the herpes zoster virus (HZV). When performed on a sample size of 1000 photos, the early detection of the HZV had an accuracy of 89.6 percent. In addition, a transfer learning strategy was utilized for the diagnosis of measles disease, which yielded a detection rate of 81.7%, a specificity rate of 97.1%, and an accuracy rate of 95.2% when utilizing the ResNet-50 model [53].

The majority of the studies that are conducted on skin diseases focus on melanoma, chickenpox, psoriasis, and measles, and very little research takes monkeypox into consideration. Data augmentation was utilized by Ali et al. [54] in order to increase the overall size of the dataset. For the purpose of categorizing monkeypox and other diseases, a threefold cross-validation experiment was prepared, and many pretrained deep learning models were put to use. Additionally, an ensemble model consisting of all three models was generated. The ensemble system achieved an accuracy of 79.26% (±1.05%), whereas VGG16 earned an accuracy of 81.48% (±6.87%) and ResNet50 achieved an accuracy of 82.96% (±4.57%). The best overall accuracy was attained by ResNet50, which was 82.96% (±4.57%). Deep artificial intelligence models offer considerable potential in diagnosing monkeypox from digital skin photographs, according to feasibility research that used state-of-the-art AI deep models to analyze skin images for the purpose of monkeypox detection. The study had precision of 85%. To train deep models, however, one needs training samples that are significantly bigger if one wants to achieve more reliable detection power [55].

Sitaula and Shahi [56] analyzed and contrasted 13 distinct pretrained DL models for the identification of the monkeypox virus. After the models were fine-tuned by the addition of universal custom layers to each of them, the results were compared utilizing four different performance criteria. In order to achieve a higher level of overall performance, the best-performing DL models were selected, and then those models were “ensembled” by using majority voting on the probabilistic outputs that were acquired from those models. In their study [57], Akin et al. compared the performance of several different image classification networks. They found that the MobileNet V2 model achieved the best performance overall, with an accuracy score of 98.25%, a sensitivity score of 96.55%, a specificity score of 100.00%, and an F1 score of 98.25%. Explainable artificial intelligence was created with the help of this model. As a result, the classification model’s judgment regarding classification was expounded upon.

Although the models for diagnosing monkeypox have been hopeful, they have increased in difficulty, making classical approaches so ineffective that they cannot solve certain NP-hard problems. Thus, we sought solutions to classic optimization methods’ issues. Metaheuristics are widely used in continuous optimization due to their robust and reliable performance, global search, lack of prerequisites for a continuously differentiable or differentiable objective function, simple implementation, implicit parallelism, and lack of specific information about the problem to be solved. This work offers machine learning techniques for effectively predicting monkeypox diseases compared with the classical approaches. A new Al-Biruni Earth radius (BER) optimization-based stochastic fractal search (BERSFS) algorithm is presented to raise the tested dataset’s prediction accuracy.

## 3. Materials and Methods

### 3.1. Convolutional Neural Network (CNN)

Convolutional neural networks are effective deep learning approaches. Although they can be used for text, signals, and other continuous data types, image inputs are where they shine the brightest when it comes to their applicability. Instead of being connected to one another, as neurons in traditional neural networks are, neurons in a convolutional layer are connected to the sub-regions of the layer that came before it. Neurons in a CNN create outputs that are spatially correlated because the sub-regions they represent may overlap. In contrast, neurons in fully connected neural networks provide outputs that are independent of one another. In addition, the number of parameters in a typical neural network can rapidly increase in response to an increase in the amount of inputs. A CNN, on the other hand, has a lower total number of network parameters since it has fewer connections, shared weights, and downsampling [29].

A CNN comprises several layers: the convolutional, max-pooling, and fully connected layers. The CNN is made up of two phases: a phase for the extraction of features followed by a phase for the classification of binary options. The proposed CNN features three different convolutional subsections. Every segment is comprised of multiple fundamental units. The fundamental unit is made up of three layers: the convolution layer, the ReLu activation layer, and the batch normalization layer. After each block of convolution, a pooling layer is formed. The sections that use convolution are used as feature extractors. The classification process is carried out by the three most recent layers: the average pooling layer, the fully connected layer, and the softmax layer. The number of basic blocks is a tunable hyperparameter that, when set, influences the architecture of the CNN [58]. The hyperparameters of the CNN are optimized with the help of Al-Biruni Earth radius (BER) optimization-based stochastic fractal search (BERSFS) algorithm, which is used to locate the most effective design.

### 3.2. Al-Biruni Earth Radius (BER) Algorithm

The Al-Biruni Earth radius (BER) optimizes the situation by segmenting the population into groups that are focused on exploration and exploitation. A dynamic process of switching up the composition of agent subgroups is used to strike a healthy balance between exploitative and exploratory activities. The population is split between exploration (which makes up 70%) and exploitation (30%). In order to raise the overall global average fitness level of both the exploration and exploitation groups, the number of agents in each of those groups has been increased. The exploration team uses mathematics to search for potentially fruitful areas in the immediate vicinity of their current location. This is accomplished by searching through the available possibilities over and over again for one that has a better degree of fitness [30].

The goal of optimization algorithms is to determine the best solution given a set of constraints. When using the BER, each member of the population is represented by a vector S. This vector takes the form $S=S_{1},S_{2},\ldots ,S_{d}\in R$, where $S_{d}$ is the size of the search space and *d* is the optimization parameter or feature. It has been recommended that the fitness function *F* be used in order to evaluate the performance of an individual up to a specific point. During these phases of optimization, populations are searched in search of a fitness-optimal vector $S^{*}$. The process begins with selecting a sample of the population at random (solutions). In order for the BER to optimize, it first requires the fitness function, as well as the lower and upper limits for each solution, the dimension, and the size of the population. Algorithm 1 provides a visual representation of the BER optimization algorithm.
**Algorithm 1 **AL-Biruni Earth radius (BER) algorithm1:**Initialize** BER population $S_{i}\left(i=1,2,\ldots ,d\right)$ with size *d*, iterations $T_{max}$, fitness function $F_{n}$, t=1, BER parameters2:**Calculate** fitness function $F_{n}$ for each $S_{i}$3:**Find** best solution as $S^{*}$4:**while**$t\leq T_{max}$**do**5:   **for** ($i=1:i<n_{1}+1$) **do**6:     **Update** $r_{1}=h_{1}\frac{cos(x)}{1-cos(x)}$, $r_{2}=h_{2}\frac{cos(x)}{1-cos(x)}$7:     **Update** positions to move toward best solution as in Equation (Equation 1)8:   **end for**9:   **for** ($i=1:i<n_{2}+1$) **do**10:     **Update** $r=h\frac{cos(x)}{1-cos(x)}$, $r_{3}=h_{3}\frac{cos(x)}{1-cos(x)}$11:     **Update** positions Elitism of best solution as in Equation (Equation 2)12:     **Update** positions Investigating area around best solution as in Equation (Equation 3)13:     **Compare** S(t+1) and $S^{\prime }\left(t+1\right)$ to select best solution $S^{*}$14:     **if** The best fitness has remained the same over the past two editions. **then**15:         **Mutate** solution as in Equation (Equation 4)16:     **end if**17:   **end for**18:   **Update** the fitness function $F_{n}$ for each $S_{i}$19:   **Find** best solution as $S^{*}$20:   **Update** BER parameters, t=t+121:**end while**22:**Return**$S^{*}$

#### 3.2.1. Exploration Operation

As will be explained in further detail below, exploration is the process that is responsible for both identifying interesting regions of the search space and avoiding local optimum stagnation through forward progress toward the optimal solution. Exploration is also responsible for determining whether or not a solution is optimal.

In order to get closer to the optimal solution, the lone explorer in the group will use this method to search for promising new regions to investigate in the near area surrounding where they are now located. To accomplish this, one must search through the numerous available options in the surrounding area in an attempt to choose one that is superior (in terms of physical health) to the others. In order to accomplish this, the research conducted by the BER makes use of the following equations:(1)$$S\left(t+1\right)=S\left(t\right)+D\left(2r_{2}-1\right),D=r_{1}\left(S\left(t\right)-1\right)$$
where S(t) is the solution vector at iteration *t* and D is the diameter of the circle in which the search agent will explore for promising areas, while 0<x≤180 and *h* is a number that is arbitrarily chosen from the range [0,2]. Both $r_{1}$ and $r_{2}$ are examples of coefficient vectors, and their values may be calculated using the formula: $r=h\frac{cos(x)}{1-cos(x)}$.

#### 3.2.2. Exploitation Operation

The group in charge of exploiting opportunities has the duty to improve upon the solutions that are currently in place. At the conclusion of each iteration, the BER identifies the participants who have achieved the highest levels of physical fitness and awards them accordingly. The purpose of exploitation is accomplished by the BER through the employment of two unique approaches, both of which are explained here.

Moving in the direction of the optimal solution using the following equation, we can make progress in the direction of the optimal solution:(2)$$S\left(t+1\right)=r^{2}\left(S\left(t\right)+D\right),D=r_{3}\left(L\left(t\right)-S\left(t\right)\right)$$
where $r_{3}$ is a random vector that is created using the formula $r=h\frac{cos(x)}{1-cos(x)}$, which regulates the movement steps toward the best solution, S(t) is the solution vector at iteration *t*, L(t) is the best solution vector, and D refers to the distance vector.

For investigating the region surrounding the optimal solution, the territory surrounding the correct response is the most interesting potential option (leader). Because of this, some people try to find ways to enhance things by investigating possibilities that are somewhat near to the best option. In order to carry out the process described above, the BER makes use of the equation shown below:(3)$$S^{\prime }\left(t+1\right)=r\left(S^{*}\left(t\right)+k\right),k=1+\frac{2\times t^{2}}{Max_{iter}^{2}}$$
where $S^{*}\left(t\right)$ is the best solution. The best solution for $S^{*}$ can be selected by comparing S(t+1) and $S^{\prime }\left(t+1\right)$. The solution will be modified according to the following equation if the best fitness has not changed throughout the course of the previous two iterations:(4)$$S\left(t+1\right)=k*z^{2}-h\frac{cos(x)}{1-cos(x)}$$
where *z* is a random number in the range [0,1].

#### 3.2.3. Selection of the Best Solution

In order to ensure that the solutions have a high level of quality, the BER chooses the one that is going to be the most effective for usage in the subsequent cycle. However, due to the enhanced efficiency of the elitism technique, multi-modal functions might converge too soon. By taking a mutational approach and examining all around the members of the exploration group, the BER is able to deliver exceptional capabilities for the exploration process. The BER is able to delay convergence thanks to its powerful exploring capabilities. The BER pseudo-code can be viewed by referring to Algorithm 1 and looking at the section titles. To begin, we will provide the BER with basic information, such as the population size, mutation rate, and number of iterations. After that, the BER splits the participants up into two groups: those who undertake work that is exploratory and those who perform work that is exploitative. The BER method will automatically make adjustments to the size of each group as it works its way through the iterative process of finding the best answer. Each team completes its tasks using two different approaches. In order to ensure a diverse and in-depth inquiry, the BER will rearrange the order of the responses whenever it performs an iteration. For instance, a solution that was part of the exploration group in the previous iteration may migrate to the exploitation group in the iteration after that. Because of the exclusive nature of the BER’s selection procedure, the leader will not be changed while the process is ongoing.

### 3.3. Stochastic Fractal Search (SFS) Algorithm

It is possible for random fractals to motivate a metaheuristic algorithm by making use of the time and accuracy of the conventional fractal technique. It is possible for a particle to have electrical potential energy, where each particle has the ability to diffuse. Other random particles can be created where the energy of the original particle is distributed among the new particles, and in each generation, the best particles are retained while the remaining particles are discarded. These three guidelines are the foundation of the fundamental fractal search (FS) procedure, which is utilized to locate a remedy for a specific issue that has been presented. The stochastic fractal search, also known as SFS, is a type of algorithm that was designed with the fractal paradigm as its foundation [59,60]. By utilizing three different update mechanisms—one for diffusion, one for the first update, and one for the second update—SFS is able to go around the limitations of FS [61,62]. The diffusion phase of the SFS algorithm consists of a series of Gaussian walks that are carried out around the best possible solution (best particle).

In the DLA growth process, the Gaussian distribution approach is used for random walk. This is so that new particles can be generated based on the diffusion mechanism of the SFS. A list of walks that were formed throughout the process of diffusion in accordance with the optimal solution $S^{*}\left(t\right)$ is created. The expression can be calculated as follows:(5)$$S^{\prime }{}^{*}\left(t+1\right)=Gaussian\left(\mu_{S^{*}},\sigma \right)+\left(\eta \times S^{*}\left(t\right)-\eta^{\prime }\times P_{i}\right)$$
where the updated best solution is represented by the symbol $S^{\prime }{}^{*}\left(t+1\right)$. The parameters of η and $\eta^{\prime }$ are random numbers ∈[0,1]. The value represented by $P_{i}$ is the location of the *i*th point in the group that surrounds the point. Since there are fewer generations around the optimal solution, the value of $\mu_{S^{*}}$ is equivalent to $\left|S^{*}\right|$, and the value of σ is equivalent to $\left|P_{i}-S^{*}\right|$. This boosts the potential of exploration in the proposed BER, which is based on the diffusion process of the SFS algorithm, so that it can find the optimal solution more quickly. Algorithm 2 shows the SFS algorithm step by step.
**Algorithm 2 **Stochastic fractal search (SFS) algorithm1:**Initialize** SFS population $S_{i}\left(i=1,2,\ldots ,d\right)$ with size *d*, iterations $T_{max}$, fitness function $F_{n}$, t=1, SFS parameters2:**Calculate** fitness function $F_{n}$ for each $S_{i}$3:**Find** best solution as $S^{*}$4:**while**$t\leq T_{max}$**do**5:     **for** (i=1:i<n+1) **do**6:        **Calculate** updated best solution as in Equation (Equation 5)7:     **end for**8:     **Update** the fitness function $F_{n}$ for each $S_{i}$9:     **Find** best solution as $S^{*}$10:   **Update** SFS parameters, t=t+111:**end while**12:**Return**$S^{*}$

### 3.4. Proposed BERSFS Algorithm

A step-by-step explanation of the proposed AL-Biruni Earth radius stochastic fractal search (BERSFS) algorithm can be found in Algorithm 3. The BERSFS algorithm mitigates the drawbacks of the BER and SFS algorithms while capitalizing on the benefits of both in order to produce the optimal answer for the whole world. The first step in the method is to find the starting positions of d preset agents using the notation $S_{i}\left(i=1,2,\ldots ,d\right)$. This also defines the parameters for the BER algorithm and the SFS algorithm, as well as the maximum number of allowable iterations for the execution process, which is denoted by $T_{max}$. A value that is completely unpredictable and falls anywhere between 0 and 1 is referred to as ${rand}_{BERSFS}$. The BER equations will be used by the BERSFS algorithm to determine how the positions of the agents should be updated if ${rand}_{BERSFS}>0.5$. If ${rand}_{BERSFS}\leq 0.5$, then the SFS equations will be used by the BERSFS algorithm to guide the process of updating the positions of the agents.

The following is an expression of the computational complexity that the BERSFS algorithm presents in this paper. For iterations $T_{max}$ and *n* number of agents, one definition of the complexity is as follows:Initialize the parameters of the BERSFS algorithm: *O*(1);Calculate $F_{n}$ for each agent $S_{i}$: *O*(*n*);Obtain the best agent $S^{*}$: *O* (*n*);Update positions to head toward the best solution: *O*($T_{max}\times n$);Update positions for the elitism of the best solution: *O*($T_{max}\times n$);Update the positions for investigating the area around the best solution: *O*($T_{max}\times n$);Mutate the solution: *O*($T_{max}$);Calculate the updated best solution: *O*($T_{max}\times n$);Calculate $F_{n}$ for each agent $S_{i}$: *O*($T_{max}$);Update the BERSFS parameters: *O*($T_{max}$);Obtain the best agent $S^{*}$: *O*($T_{max}$);Obtain the best agent $S^{*}$: *O*(1).
**Algorithm 3 **Proposed BERSFS algorithm1:**Initialize** BERSFS population $S_{i}\left(i=1,2,\ldots ,d\right)$ with size *d*, iterations $T_{max}$, fitness function $F_{n}$, t=1, BERSFS parameters2:**Calculate** fitness function $F_{n}$ for each $S_{i}$3:**Find** best solution as $S^{*}$4:**while**$t\leq T_{max}$**do**5:   **if** (${rand}_{BERSFS}>0.5$) **then**6:       **for** ($i=1:i<n_{1}+1$) **do**7:           **Update** $r_{1}=h_{1}\frac{cos(x)}{1-cos(x)}$, $r_{2}=h_{2}\frac{cos(x)}{1-cos(x)}$8:           **Calculate** $D=r_{1}\left(S\left(t\right)-1\right)$9:           **Update** positions to head toward best solution as           $S\left(t+1\right)=S\left(t\right)+D\left(2r_{2}-1\right)$10:     **end for**11:     **for** ($i=1:i<n_{2}+1$) **do**12:        **Update** $r=h\frac{cos(x)}{1-cos(x)}$, $r_{3}=h_{3}\frac{cos(x)}{1-cos(x)}$13:        **Calculate** $D=r_{3}\left(L\left(t\right)-S\left(t\right)\right)$14:        **Update** positions Elitism of best solution as        $S\left(t+1\right)=r^{2}\left(S\left(t\right)+D\right)$15:        **Calculate** $k=1+\frac{2\times t^{2}}{Max_{iter}^{2}}$16:        **Update** positions Investigating area around best solution as        $S^{\prime }\left(t+1\right)=r\left(S^{*}\left(t\right)+k\right)$17:        **Compare** S(t+1) and $S^{\prime }\left(t+1\right)$ to select best solution $S^{*}$18:        **if** best fitness is not changed for last two iterations **then**19:            **Mutate** solution as $S\left(t+1\right)=k*z^{2}-h\frac{cos(x)}{1-cos(x)}$20:        **end if**21:     **end for**22:   **else**23:     **for** (i=1:i<n+1) **do**24:        **Calculate** updated best solution as        $S^{\prime }{}^{*}\left(t+1\right)=Gaussian\left(\mu_{S^{*}},\sigma \right)+\left(\eta \times S^{*}\left(t\right)-\eta^{\prime }\times P_{i}\right)$25:     **end for**26:   **end if**27:   **Update** the fitness function $F_{n}$ for each $S_{i}$28:   **Find** best solution as $S^{*}$29:   **Update** BERSFS parameters, t=t+130:**end while**31:**Return**$S^{*}$

According to the preceding investigation of the BERSFS algorithm, the level of computational complexity was determined to be *O*($T_{max}\times n$).

## 4. Experimental Results

This section devotes its entirety to providing an in-depth analysis of the investigation’s results. There were three distinct environments in which the studies were conducted. In the first scenario, the BERSFS algorithm’s capabilities for the dataset under testing were covered in comparison with those of basic models. In the second situation, however, the algorithm’s classification capabilities were demonstrated in comparison with those of deep learning models. In the final example, the classification skills of the BERSFS algorithm were contrasted with those of state-of-the-art optimization techniques that are based on CNNs. Table 1 shows the parameters for the BERSFS algorithm’s configuration. The parameters of the experimental CNN configuration are presented in Table 2. These are the values that will be used in this experiment.

### 4.1. Dataset Description

An openly accessible dataset on Kaggle [63] served as the foundation for the experiments that were carried out as part of this body of work. Concern has been raised among medical professionals all over the world in response to the most recent monkeypox outbreak. In order to accomplish this goal, a novel skin image-based dataset dedicated to the diagnosis of monkeypox disease was developed. There were a total of 293 photographs of normal cases, 279 images of monkeypox cases, 107 images of chickenpox patients, and 91 images of measles cases included in this collection. Sources based on the internet were used to compile all of the image classes. The Department of Computer Science and Engineering at Islamic University, which is located in Kushtia-7003, Bangladesh, was responsible for the development of the full dataset. Figure 1 displays several example images taken from the dataset. During the preprocessing phase, each of the images that were included in the dataset for each of the four separate types—monkeypox, chickenpox, measles, and normal—were scaled and normalized to have dimensions of 227×227. This step took place before any of the other processing steps. This was performed before the images were incorporated into the model being proposed.

### 4.2. Performance Metrics

The models can be evaluated with a number of different performance indicators, such as accuracy, precision, F1 score, recall, specificity, and sensitivity. The number of cases that were correctly diagnosed as being defective is represented by the letter “TP”, which stands for “true positive”. The number of cases that were wrongly recognized as defective is referred to as the “false positive”, and its abbreviation is “FP”. In addition, “false negative” (abbreviated as “FN”) refers to the number of instances that were wrongly categorized as non-defective and shows the total number of these occurrences. The abbreviation “TN” stands for “true negative”, which refers to the number of instances that were accurately identified as being free of defects. The definitions of the measures can be found in Table 3.

### 4.3. Comparison with Basic Models

In the trials, the classification results of the presented BERSFS algorithm, which is based on the CNN classifier, are discussed. There were 20 runs, and 80 iterations with 10 agents were used, as shown in Table 1. The fundamental classifiers of the decision tree (DT) Regressor [36], K-nearest neighbors (K-NN) [37], and support vector machine (SVM) [38] models were initially evaluated and contrasted with the suggested BERSFS algorithm-based CNN technique before being applied to the dataset that would be examined.

The effectiveness of the proposed method, which is referred to as BERSFS-CNN, was evaluated in comparison to the widely utilized classifiers CNN, SVM-Linear, k-NN, and DT. The results of this comparison are presented in Table 4. As a result of the BERSFS approach improving the performance of the CNN after tweaking its hyperparameters, the BERSFS-CNN model provided the best results when employed as a classifier, with an accuracy of 0.9883. The simple CNN model achieved the second-best accuracy with a score of 0.9337, followed by the SVM-Linear model with a score of 0.9213 and then the K-NN model with a score of 0.8777. The DT model achieved the lowest accuracy at 0.8510. This was as a result of the fact that the performance of the CNN was enhanced by the algorithm that was proposed.

The results of the Wilcoxon signed-rank test and the analysis of variance (based on 10 runs and 80 iterations for 10 agents) results are included in Table 5 and Table 6, respectively. This statistical testing, which had a *p* value that is less than 0.05, revealed that there was a significant difference between the outcomes of using the suggested BERSFS-CNN classifier and the outcomes of using alternative methods. Figure 2 shows the homoscedasticity graphs, heat maps, residual plots, and QQ plots of the BERSFS-CNN and other basic models. The figure confirms the performance of the proposed model.

### 4.4. Comparison with Deep Learning Models

The deep learning models of AlexNet [39], VGG19Net [40], GoogLeNet [41], and ResNet-50 [42] were used as benchmarks for the BERSFS-CNN algorithm that was developed. The performance metrics used in this part are listed in Table 3. The effectiveness of the proposed method, which is referred to as BERSFS-CNN, was evaluated in comparison to the widely utilized deep learning models. The results of this comparison are presented in Table 7. When put to use as a classifier, the BERSFS-CNN model produced the highest-quality results, achieving an accuracy of 0.9883. This was due to the fact that the BERSFS technique resulted in an improvement in the CNN’s performance after the hyperparameters were tuned. With a score of 0.9459, the AlexNet model achieved the second-best accuracy, followed by the GoogLeNet model with a score of 0.9351 and the VGG19Net model with a score of 0.9280. The ResNet-50 model achieved the lowest level of accuracy possible at 0.9208. This was due to the improved performance of the CNN, which was the driving force for the decision to make the shift.

Table 8 and Table 9 present the findings of the Wilcoxon signed-rank test and the analysis of variance, respectively. These findings were based on 10 runs of the simulation and 80 iterations for 10 different agents. This statistical test, which had a *p* value that is less than 0.05, demonstrated that there was a significant difference between the results of using the suggested BERSFS-CNN classifier and the results of applying alternative approaches. The *p* value for this test was less than 0.05. Figure 3 depicts the homoscedasticity graphs, heat maps, residual plots, and QQ plots of the BERSFS-CNN, in addition to those of the other deep learning models. Heat maps provide a visual representation of the temperature of an area. The effectiveness of the model that was suggested can be shown to be proven by looking at the figure.

### 4.5. Comparison with Optimization-Based Models

The classification results of the suggested algorithm are compared with the results of other CNN-based models, including the BER-CNN [30], SFS-CNN [32], WOA-CNN [35], GWO-CNN [34], and PSO-CNN [33], in order to demonstrate the effectiveness of the provided method. Table 10 shows the parameter settings for the methods being compared. The performance metrics used in this part are listed in Table 3.

Table 11 displays the classification outcomes of the proposed and contrasted techniques that are based on optimizing the CNN model’s parameters. The BERSFS-CNN model generated results with an accuracy value that could reach a maximum of 0.9883. The results reveal that the model was superior to cutting-edge classifier models that were constructed using the CNN technique. The BER-CNN-based model produced the second-best classification results with an accuracy value of 0.9759, followed by the SFS-CNN-based model with a score of 0.972, the PSO-CNN-based model with a score of (0.968), the GWO-CNN-based model with a score of 0.9636, and the WOA-CNN-based model with a score of 0.9598, which produced the results that were the least accurate.

Table 12, Table 13 and Table 14 contain statistical descriptions of the provided BERSFS-CNN classifier and the comparative classifiers, respectively, as well as the results of the ANOVA and Wilcoxon signed-rank tests based on 10 runs and 80 iterations for 10 agents. This statistical testing indicates the significant difference between the suggested BERSFS-CNN classifier’s outcomes and those of other algorithms, with a *p* value of less than 0.05. In addition to the homoscedasticity graphs, heat maps, residual plots, and QQ plots of the other fundamental models that are displayed in Figure 4, the BERSFS-CNN also contains its very own versions of each of these sorts of plots. When one looks at the graphic Figure 4, it is very clear that the usefulness of the model that was provided is proven.

The box plot in Figure 5 illustrates the accuracy of the proposed BERSFS-CNN and compared approaches. The BERSFS-CNN-based optimization algorithm’s highest accuracy results are shown in the graph below. The accuracy histogram for the algorithms presented and compared is displayed in Figure 6, and this is based on the number of values with the bin center range 0.938–0.968, which attests to the stability of the suggested approach. Figure 7 and Figure 8 show the ROC curves of the proposed BERSFS-CNN algorithm versus the BER-CNN and WOA-CNN algorithms, respectively. The ROC curves confirm the stability of the proposed BERSFS-CNN algorithm.

## 5. Conclusions

To fine-tune the deep convolutional neural network (DCNN) layers, we offer a new framework that uses Al-Biruni Earth radius (BER) optimization-based stochastic fractal search (BERSFS). In the first stage of the proposed method, deep CNN-based models are utilized in order to acquire knowledge regarding the embedding of input images into Euclidean space. In the second stage, we calculate the distance between pairs of images in Euclidean space using an improved classification model that is based on the triplet loss function. This allows us to learn features that may be used to differentiate between different cases, including monkeypox cases. The method that has been proposed makes use of photographs of human facial skin illnesses that were collected from a hospital in Africa. The experimental findings of the study provide evidence that the proposed framework is effective, outperforming a large number of other studies that were conducted previously on skin disease issues. On the other hand, statistical experiments using the Wilcoxon and analysis of variance (ANOVA) tests were carried out in order to assess the proposed method with regard to its efficiency and consistency. When compared with other optimization algorithms and machine learning models, the findings that were recorded provided conclusive evidence that the method that was provided was superior. 

## Figures and Tables

**Figure 1 diagnostics-12-02892-f001:**
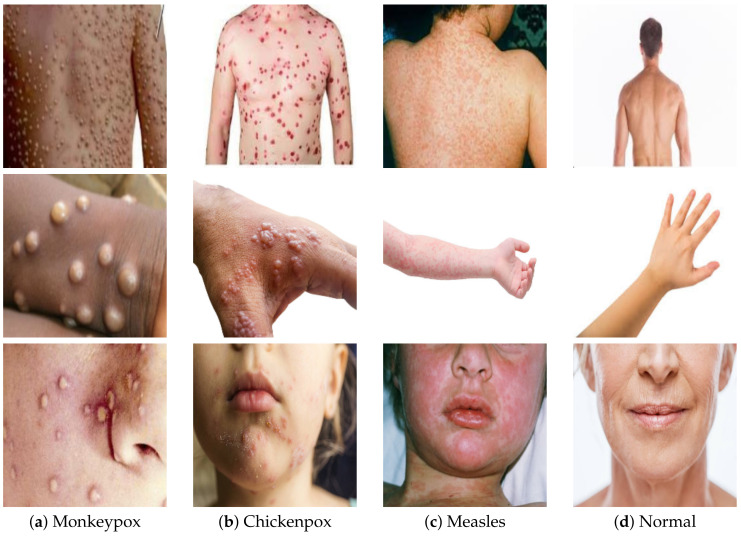
Samples from the Monkeypox Skin Images Dataset (MSID) [63]: (**a**) monkeypox cases, (**b**) chickenpox cases, (**c**) measles cases, and (**d**) normal cases.

**Figure 2 diagnostics-12-02892-f002:**
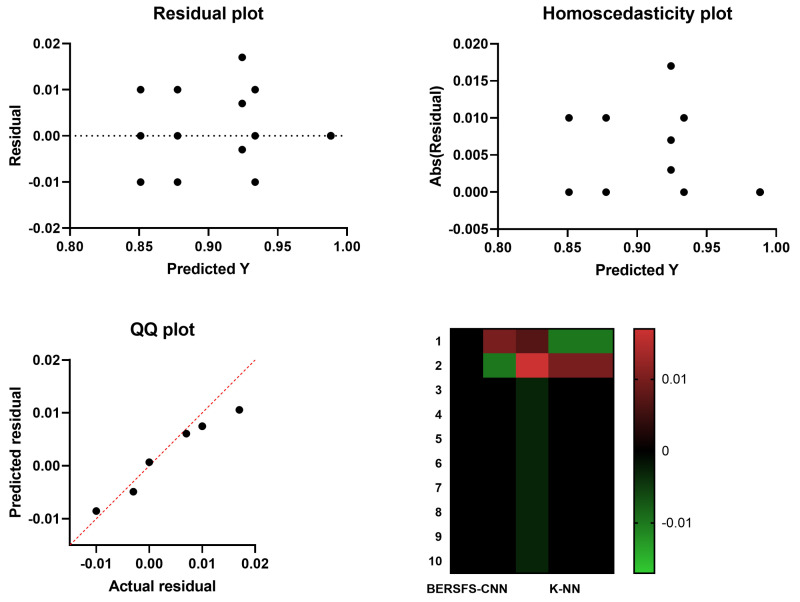
The homoscedasticity graphs, heat maps, residual plots, and QQ plots of the BERSFS and other basic models.

**Figure 3 diagnostics-12-02892-f003:**
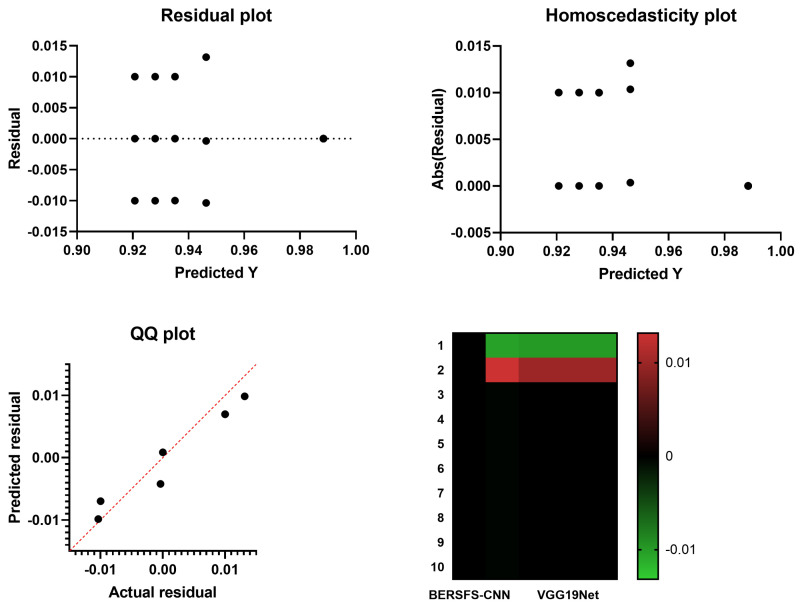
The homoscedasticity graphs, heat maps, residual plots, and QQ plots of the BERSFS-CNN and other deep learning models.

**Figure 4 diagnostics-12-02892-f004:**
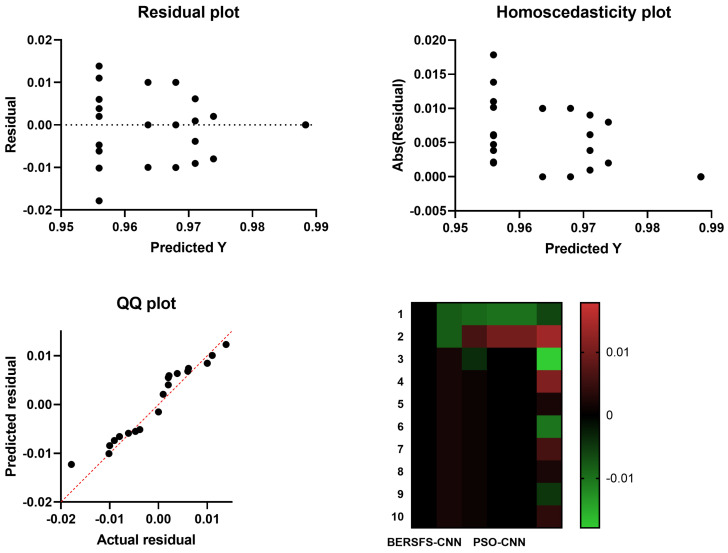
The homoscedasticity graphs, heat maps, residual plots, and QQ plots of the BERSFS-CNN and other optimization-based models.

**Figure 5 diagnostics-12-02892-f005:**
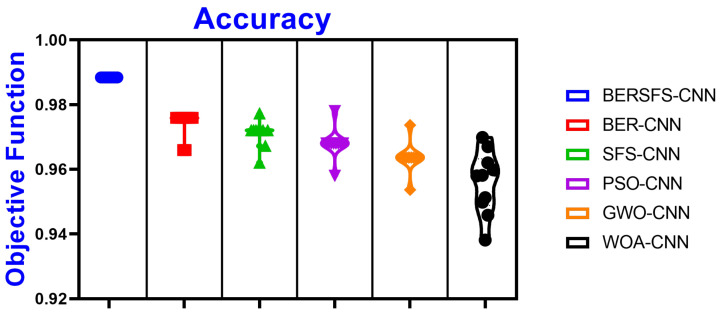
The box plot of the accuracy of the proposed BERSFS-CNN and comparison approaches.

**Figure 6 diagnostics-12-02892-f006:**
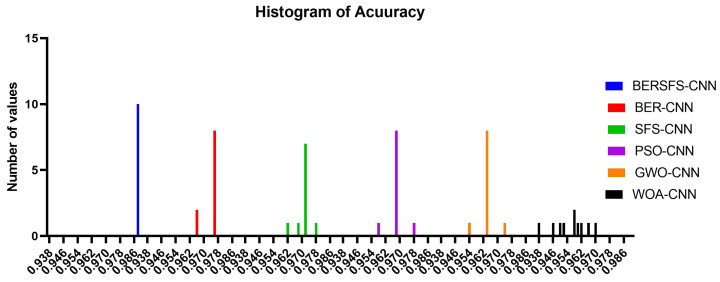
The accuracy histogram for the algorithms presented and compared with number of values in the bin center range 0.938–0.968.

**Figure 7 diagnostics-12-02892-f007:**
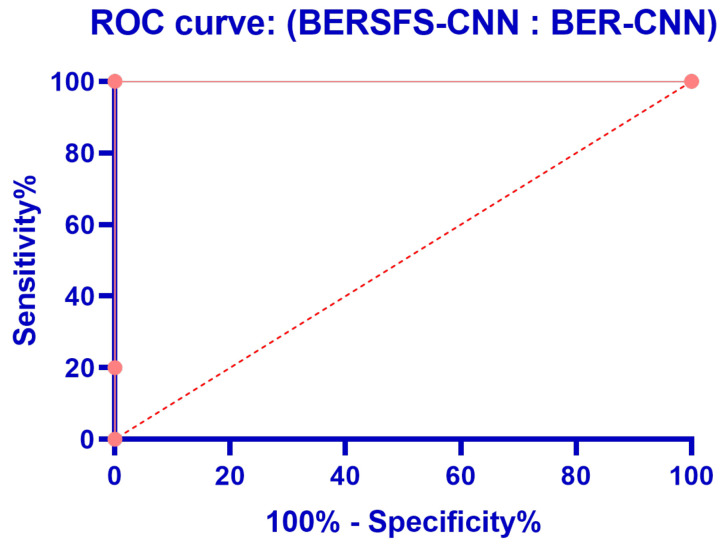
ROC curve of the proposed BERSFS-CNN algorithm versus the BER-CNN algorithm.

**Figure 8 diagnostics-12-02892-f008:**
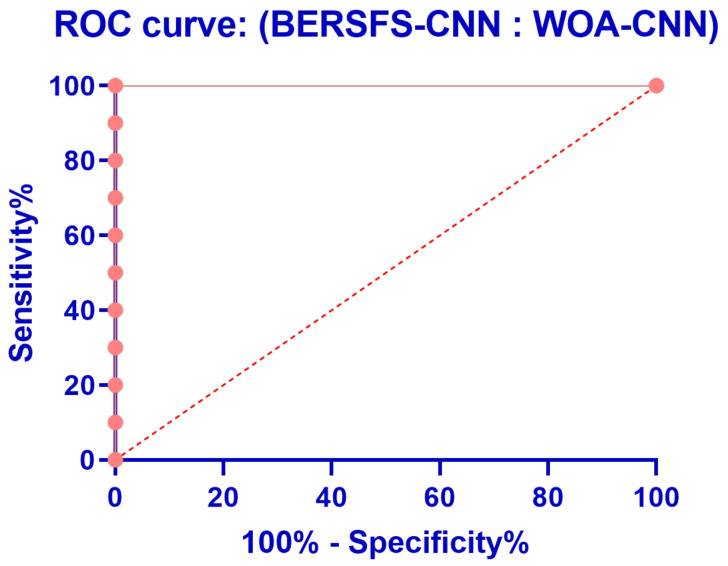
ROC curve of the proposed BERSFS-CNN algorithm versus the WOA-CNN algorithm.

**Table 1 diagnostics-12-02892-t001:** Parameters for BERSFS algorithm configuration.

Parameter	Value
Number of Agents	10
Number of Iterations	80
Number of Repetitions	20
η	∈[0, 1]
$\eta^{\prime }$	∈[0, 1]
Mutation probability	0.5
Exploration percentage	70
K (decreases from 2 to 0)	1

**Table 2 diagnostics-12-02892-t002:** Parameters of the experimental CNN configuration.

Parameter	Value
**CNN training options (Default)** Momentum Learn RateDropFactor L2Regularization LearnRateDropPeriod GradientThreshold GradientThresholdMethod ValidationData VerboseFrequency ValidationPatience ValidationFrequency ResetInputNormalization **CNN training options (Custom)** InitialLearnRate ExecutionEnvironment BatchSize MaxEpochs Verbose Shuffle LearnRateSchedule Optimizer	0.9000 0.1000 1.0000 × 10${}^{-4}$ 10 Inf l2norm imds 50 Inf 50 1 0.001 gpu 32 40 0 every-epoch piecwise BERSFS

**Table 3 diagnostics-12-02892-t003:** Performance metrics used in CNN.

No. Calculation	Metrics
Accuracy	$\frac{TP+TN}{TP+TN+FP+FN}$
Sensitivity	$\frac{TP}{TP+FN}$
Specificity	$\frac{TN}{TN+FP}$
Precision (PPV)	$\frac{TP}{TP+FP}$
Negative Predictive Value (NPV)	$\frac{TN}{TN+FN}$
F1 Score	$2\times \frac{PPV\times TPR}{PPV+TPR}$

**Table 4 diagnostics-12-02892-t004:** Results of comparisons between proposed BERSFS-CNN and basic models.

	Accuracy	Sensitivity (TRP)	Specificity (TNP)	*p* Value (PPV)	N Value (NPV)	F1 Score
BERSFS-CNN	0.9883	0.8571	0.9921	0.7595	0.9959	0.8054
CNN	0.9337	0.7500	0.9693	0.8257	0.9524	0.7860
SVM-Linear	0.9213	0.8571	0.9231	0.2308	0.9959	0.3636
K-NN	0.8777	0.8000	0.9132	0.8081	0.9091	0.8040
DT	0.8510	0.7273	0.9132	0.8081	0.8696	0.7656

**Table 5 diagnostics-12-02892-t005:** Results of comparisons between proposed classifiers and those obtained from ANOVA.

	SS	DF	MS	F (DFn, DFd)	*p* Value
Treatment (between columns)	0.1129	4	0.0282	F (4, 45) = 1258	*p* < 0.0001
Residual (within columns)	0.0010	45	$2.24\times 10^{-5}$	-	-
Total	0.1139	49	-	-	-

**Table 6 diagnostics-12-02892-t006:** Test results for the Wilcoxon signed-rank test of the BERSFS-CNN model and basic classifiers.

	BERSFS-CNN	CNN	SVM-Linear	K-NN	DT
Theoretical median	0	0	0	0	0
Actual median	0.9883	0.9337	0.9213	0.8777	0.8511
Number of values	10	10	10	10	10
Wilcoxon signed-rank test					
Sum of signed ranks (W)	55	55	55	55	55
Sum of positive ranks	55	55	55	55	55
Sum of negative ranks	0	0	0	0	0
*p* value (two-tailed)	0.002	0.002	0.002	0.002	0.002
Exact or estimate?	Exact	Exact	Exact	Exact	Exact
Significant (alpha = 0.05)?	Yes	Yes	Yes	Yes	Yes
How big is the discrepancy?					
Discrepancy	0.9883	0.9337	0.9213	0.8777	0.8511

**Table 7 diagnostics-12-02892-t007:** Results of comparisons between proposed BERSFS-CNN and deep learning models.

	Accuracy	Sensitivity (TRP)	Specificity (TNP)	*p* Value (PPV)	N Value (NPV)	F1 Score
BERSFS-CNN	0.9883	0.8571	0.9921	0.7595	0.9959	0.8054
AlexNet	0.9459	0.7143	0.9524	0.2941	0.9917	0.4167
GoogLeNet	0.9351	0.7143	0.9412	0.2500	0.9917	0.3704
VGG19Net	0.9280	0.7143	0.9339	0.2273	0.9917	0.3448
ResNet-50	0.9208	0.6667	0.9266	0.1739	0.9917	0.2759

**Table 8 diagnostics-12-02892-t008:** Results of comparisons between proposed classifiers and those obtained from ANOVA.

	SS	DF	MS	F (DFn, DFd)	*p* Value
Treatment (between columns)	0.0284	4	0.0071	F (4, 45) = 363.3	*p* < 0.0001
Residual (within columns)	0.0009	45	$0.1959\times 10^{-4}$	-	-
Total	0.0293	49	-	-	-

**Table 9 diagnostics-12-02892-t009:** Results for the Wilcoxon signed-rank test of the BERSFS-CNN model and deep learning models.

	BERSFS-CNN	AlexNet	GoogLeNet	VGG19Net	ResNet-50
Theoretical median	0	0	0	0	0
Actual median	0.9883	0.9459	0.9351	0.928	0.9208
Number of values	10	10	10	10	10
Wilcoxon signed-rank test					
Sum of signed ranks (W)	55	55	55	55	55
Sum of positive ranks	55	55	55	55	55
Sum of negative ranks	0	0	0	0	0
*p* value (two-tailed)	0.002	0.002	0.002	0.002	0.002
Exact or estimate?	Exact	Exact	Exact	Exact	Exact
Significant (alpha = 0.05)?	Yes	Yes	Yes	Yes	Yes
How big is the discrepancy?					
Discrepancy	0.9883	0.9459	0.9351	0.928	0.9208

**Table 10 diagnostics-12-02892-t010:** Parameter settings for the methods being compared.

Algorithm	Parameter (s)	Value (s)
BER	Mutation probability	0.5
	Exploration percentage	70
	K (decreases from 2 to 0)	1
SFS	η	∈[0, 1]
	$\eta^{\prime }$	∈[0, 1]
PSO	Acceleration constants	[2, 2]
	Inertia $W_{max}$, $W_{min}$	[0.6, 0.9]
	Particles	10
	Iterations	80
GWO	*a*	2 to 0
	Iterations	80
	Wolves	10
WOA	*r*	[0, 1]
	Iterations	80
	Whales	10
	*a*	2 to 0

**Table 11 diagnostics-12-02892-t011:** Classification outcomes of the proposed and contrasted techniques that were based on optimizing the CNN model’s parameters.

	Accuracy	Sensitivity (TRP)	Specificity (TNP)	*p* Value (PPV)	N Value (NPV)	F1 Score
BERSFS-CNN	0.9883	0.8571	0.9921	0.7595	0.9959	0.8054
BER-CNN	0.9759	0.7500	0.9796	0.3750	0.9959	0.5000
SFS-CNN	0.9720	0.6000	0.9796	0.3750	0.9917	0.4615
PSO-CNN	0.9680	0.4000	0.9796	0.2857	0.9877	0.3333
GWO-CNN	0.9636	0.4000	0.9767	0.2857	0.9859	0.3333
WOA-CNN	0.9598	0.7674	0.9655	0.3976	0.9929	0.5238

**Table 12 diagnostics-12-02892-t012:** Results of comparisons between proposed classifiers and those obtained from ANOVA.

	SS	DF	MS	F (DFn, DFd)	*p* Value
Treatment (between columns)	0.0059	5	0.0012	F (5, 54) = 41.27	*p* < 0.0001
Residual (within columns)	0.0016	54	$2.88\times 10^{-5}$	-	-
Total	0.0075	59	-	-	-

**Table 13 diagnostics-12-02892-t013:** Test results for the Wilcoxon signed-rank test of the BERSFS-CNN model and optimization-based models.

	BERSFS-CNN	BER-CNN	SFS-CNN	PSO-CNN	GWO-CNN	WOA-CNN
Theoretical median	0	0	0	0	0	0
Actual median	0.9883	0.9759	0.972	0.968	0.9636	0.9581
Number of values	10	10	10	10	10	10
Wilcoxon signed-rank test						
Sum of signed ranks (W)	55	55	55	55	55	55
Sum of positive ranks	55	55	55	55	55	55
Sum of negative ranks	0	0	0	0	0	0
*p* value (two-tailed)	0.002	0.002	0.002	0.002	0.002	0.002
Exact or estimate?	Exact	Exact	Exact	Exact	Exact	Exact
Significant (alpha = 0.05)?	Yes	Yes	Yes	Yes	Yes	Yes
How big is the discrepancy?						
Discrepancy	0.9883	0.9759	0.972	0.968	0.9636	0.9581

**Table 14 diagnostics-12-02892-t014:** Statistical descriptions of the provided BERSFS-CNN classifier and the comparative classifiers.

	BERSFS-CNN	BER-CNN	SFS-CNN	PSO-CNN	GWO-CNN	WOA-CNN
Number of values	10	10	10	10	10	10
Minimum	0.9883	0.9659	0.962	0.9580	0.9536	0.9381
25th percentile	0.9883	0.9734	0.9708	0.9680	0.9636	0.9488
Median	0.9883	0.9759	0.972	0.9680	0.9636	0.9581
75th percentile	0.9883	0.9759	0.972	0.9680	0.9636	0.9632
Maximum	0.9883	0.9759	0.9772	0.9780	0.9736	0.9698
Range	0	0.0100	0.0152	0.0200	0.0200	0.0317
Mean	0.9883	0.9739	0.9710	0.9680	0.9636	0.9560
Std. deviation	0	0.0042	0.0036	0.0047	0.0047	0.0097
Std. error of mean	0	0.0013	0.0013	0.0015	0.0015	0.0031
Geometric mean	0.9883	0.9739	0.9710	0.9680	0.9636	0.9559
Geometric SD factor	1	1.0040	1.0040	1.0050	1.0050	1.0100
Sum	9.8830	9.7390	9.7100	9.6800	9.6360	9.5600

## Data Availability

Not applicable.

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
