# Peer review of "An Al-Biruni Earth Radius Optimization-Based Deep Convolutional Neural Network for Classifying Monkeypox Disease"

_diagnostics, 2022, doi:10.3390/diagnostics12112892_

Round 1

Reviewer 1 Report

1) The problem statement is not clearly defined in the abstract.

2) The literature survey seems to be a summary rather than a critical analysis.

3) More emphasis is given on optimization but nothing is mentioned about CNN.

4) There are many parameters in CNN. What parameters are you trying to optimize?

5) The architecture specifications of CNN is not given which will impact the performance measures.

6) The given images seems to be available with different resolution. What is the impact of resolution on the results?

7) The optimized and the unoptimized parameters must be shown.

8) The confusion matrix of the various classifiers must be given.

9) Which method do you think the best? Why?

Author Response

Thanks to the reviewer for this comment

Reviewer 2 Report

The manuscript entitled "Al-Biruni Earth Radius Optimization Based Deep Convolutional Neural Network for Classifying Monkeypox Diseases" shows a study aimed to create a computer-aided approach for the automated diagnosis of monkeypox diseases. Authors propose a novel framework using Al-Biruni Earth Radius (BER) optimization-based Stochastic Fractal 10 Search (BERSFS) to fine-tune the Deep Convolutional Neural Network (DCNN) layers.

The manuscript is structured in 5 sections: Introduction, Literature Review, Material and Methods, Experimental Results and Conclusions. The introduction is clear and it is easy to read the justification of the study and the key contributions of this work. The literature review is updated and very interesting on the subject of the classification of monkeypox diseases. The material and methods are well explained and the results are very justified and explained.

I only suggest some major and minor changes that should be fixed before the manuscript be considered as accepted for publication.

1. The reproducibility of the manuscript is important, so, the authors should share a GitHub or similar to download the testbeds and run the experiments.

2. It is important to unify the criteria of decimals throughout the manuscript. It's recommended to use three or four decimals, not more, in all the values exposed.

3. It is very important to have the figures very closest to the mentions in the text.

4. Some figures can be reduced to achieve point 3.

5. Figure 1 should be improved.

Author Response

Thanks to the reviewer for this comment

Round 2

Reviewer 1 Report

It can be accepted now

Author Response

Thanks for your support and nice revision notes, which made the manuscript superiority

Reviewer 2 Report

I am grateful that the authors have taken some of my suggestions into account.

However, I consider it very important that they share the code and data to verify that the values indicated in the manuscript are real. It is very serious for any applied methodology not to have sufficient tools to validate, as reviewers, that the information indicated in the manuscript is true.

For this reason, I again ask the authors to make available the experiments and data to validate the results shown in the manuscript.

Author Response

We agree with the reviewer on the importance of sharing the code to allow for the reproduction of the conducted experiments. This work is part of a large project funded by Princess Nourah bint Abdulrahman University. The authors find it crucial to get the university's approval before sharing the code with the research community. This could happen after achieving the goals of the project. Regarding the dataset employed in this work, it can be found by following this link:

https://www.kaggle.com/datasets/dipuiucse/monkeypoxskinimagedataset

Round 3

Reviewer 2 Report

After requesting the code twice from the authors and without being able to verify the reproducibility and veracity of the data presented, I cannot accept this article as a reviewer.

Authors have the possibility to share the code with reviewers privately.